# The Development of Dermal Self-Double-Emulsifying Drug Delivery Systems: Preformulation Studies as the Keys to Success

**DOI:** 10.3390/ph16101348

**Published:** 2023-09-25

**Authors:** Daniélle van Staden, Richard K. Haynes, Joe M. Viljoen

**Affiliations:** 1Faculty of Health Sciences, Centre of Excellence for Pharmaceutical Sciences (PharmacenTM), Building G16, North-West University, 11 Hoffman Street, Potchefstroom 2520, South Africa; dvanstaden711@gmail.com (D.v.S.); haynes@ust.hk (R.K.H.); 2Rural Health Research Institute, Charles Sturt University, 346 Leeds Parade, Orange, NSW 2800, Australia

**Keywords:** clofazimine, isoniazid, lipid-based drug delivery, multiple emulsion, pseudoternary phase diagram, pyrazinamide, rifampicin, self-double-emulsifying drug delivery system, self-emulsifying drug delivery system, skin

## Abstract

Self-emulsifying drug delivery systems (SEDDSs) are lipid-based systems that are superior to other lipid-based oral drug delivery systems in terms of providing drug protection against the gastrointestinal (GI) environment, inhibition of drug efflux as mediated by P-glycoprotein, enhanced lymphatic drug uptake, improved control over plasma concentration profiles of drugs, enhanced stability, and drug loading efficiency. Interest in dermal spontaneous emulsions has increased, given that systems have been reported to deliver drugs across mucus membranes, as well as the outermost layer of the skin into the underlying layers. The background and development of a double spontaneous emulsion incorporating four anti-tubercular drugs, clofazimine (CFZ), isoniazid (INH), pyrazinamide (PZY), and rifampicin (RIF), are described here. Our methods involved examination of oil miscibility, the construction of pseudoternary phase diagrams, the determination of self-emulsification performance and the emulsion stability index of primary emulsions (PEs), solubility, and isothermal micro calorimetry compatibility and examination of emulsions via microscopy. Overall, the potential of self-double-emulsifying drug delivery systems (SDEDDSs) as a dermal drug delivery vehicle is now demonstrated. The key to success here is the conduct of preformulation studies to enable the development of dermal SDEDDSs. To our knowledge, this work represents the first successful example of the production of SDEDDSs capable of incorporating four individual drugs.

## 1. Introduction

Lipid-based drug delivery systems are under intensive examination for improving the oral bioavailability of lipophilic drugs [1,2]. Self-emulsifying drug delivery systems (SEDDSs) are superior to other lipid-based systems by providing protection against the gastrointestinal (GI) environment, inhibition of drug efflux mediated by P-glycoprotein, increased lymphatic drug uptake, improved control of plasma drug concentration profiles, improved stability, and enhanced drug loading efficiency [3,4]. SEDDS are an isotropic mixture of synthetic and/or natural oils used in combination with a surfactant and co-surfactant that can spontaneously transform into an emulsion when introduced with gentle agitation into aqueous media [1,4]. The original conceptual basis of the manufacture of SEDDSs was via introduction of an isotropic mixture of oil and surface active agents into GI fluids under mild agitation, as achieved by the peristaltic movements of the GI tract, in order to establish self-emulsification [1,4]. SEDDSs have also been shown to penetrate barriers such as the mucus gel layer of the GI tract, and the outermost protective skin barrier [2,5,6].

The mucus gel layer of the GI tract comprises water (90–95%), glycoproteins, and additional minor components such as DNA, lipids, and electrolytes [5]. These components must be considered when developing formulations that have to cross this barrier [5]. Mucus glycoproteins comprise an entangled visco-elastic network that protects the underlying GI epithelium against xenobiotics and microorganisms [5]. Thus, this barrier will hinder delivery of nanosized drug delivery systems, leading to the reduced oral bioavailability of the incorporated drugs [5]. However, SEDDSs that incorporate a hydrophilic polyethylene glycol (PEG) surface layer (referred to as a PEGylated surface) are able to cross this mucus barrier [5]. Moreover, SEDDSs with a droplet size <50 nm are able to diffuse easily through the mucus network; this layer generally prohibits the entry of substances and organisms of sizes ranging between 100–200 nm [5]. Also, SEDDSs with proteolytic enzymes anchored on the surface can potentially increase mucus permeability [7]. Hence, given the hydrophilic nature of the mucus layer of the GI tract, it is noteworthy that SEDDSs are able to cross this barrier. SEDDSs also have the capacity to deliver drugs through the lipophilic barrier provided by the outermost skin layer, that is, the stratum corneum (SC) [6]. The capacity of a drug delivery vehicle to cross such divergent biological barriers is thus worthy of further investigation.

Compared to conventional drug administration routes, the dermal route is considered safe and non-invasive, and is especially useful for delivering drugs of poor aqueous solubility [8]. Two important questions arise in relation to formulation of SEDDSs: (i) can the mechanism of spontaneous emulsification be defined? (ii) is it possible to modulate the spontaneous emulsification properties of these lipid-based formulations? [9]. Notably, utilization of these emulsions as vehicles for the simultaneous delivery of multiple drugs remains a slowly growing field [10]. The simultaneous delivery of multiple drugs poses several challenges, including drug–drug interactions, and the need to select components of the emulsion that are able to dissolve and incorporate the drugs as well as the excipients [10]. Therefore, although SEDDSs are valuable topical/transdermal drug delivery systems, there do remain certain limitations [4,6]. Nevertheless, the further investigation of self-double-emulsifying drug delivery systems (SDEDDSs) is required in order to develop dermal emulsions as multidrug delivery vehicles [11].

## 2. Results and Discussion

### 2.1. Oil Immiscibility Studies

All experiments were carried out in triplicate, and the results are expressed in Figure 1 as mean ± SD (*n* = 3) values for the different oils tested. No external oil phase mixed with beeswax or linseed oil displayed clear separation. Further, no separation was detected between shea butter and any internal oil phase. Therefore, Transcutol^®^:PEG (9:1) was selected as the only suitable internal oil phase. Clear separation between Transcutol^®^:PEG (9:1) and natural oils was detected in the following order: evening primrose oil (EPO) > olive oil (OLV) > avocado oil (AVO) > palm oil (PALM) > safflower oil (SAFF).

These results can be explained by the polar natures of Transcutol^®^ and PEG due to their relatively high dielectric constants and hydrogen bonding [2,12]. Transcutol^®^ also known as diethylene glycol monoethyl ether, is a liquid well known to the cosmetic industry due to its favorable solubility properties that improve the solubilization of lipophilic and hydrophilic drugs [13]. Moreover, Transcutol^®^ is frequently included as a co-surfactant and skin penetration enhancer [13]. The addition of Transcutol^®^ to a dermal formulation with an aqueous component will influence the ionization equilibrium of the incorporated drugs, should they be capable of undergoing ionization [12]. This solvent effect occurs when a liquid with a dielectric constant lower than that of water is added to an aqueous solution of a drug. The consequence of solvating drugs with a solvent of a lower dielectric constant suppresses the extent of ionization and enhances ion pair formation at the expense of forming solvent-separated ion pairs [12]. Therefore, the larger amount of neutral, unionized drug and improved formation of ion pairs induced by the addition of Transcutol^®^ will increase the quantity of un-ionized drug able to cross the skin barrier [12].

It is reported that inclusion of both water and Transcutol^®^ in dermal formulations is advantageous, as the absence of water decreases the thermodynamic driving force needed to enable penetration of drug into the skin [12]. The addition of water permits a drug to precipitate from solution, and therefore enhances the driving force required to establish dermal flux [12]. This is especially true for lipophilic drugs, since lipophilic drugs may tend to remain in a lipophilic vehicle or create a reservoir effect where the drug remains in the lipophilic SC instead of entering the underlying hydrophilic layers of the skin [4,6]. Hence, it can be argued that it would be better to include one of the natural oils as the internal oil phase so as to enable Transcutol^®^ to fulfil the role of the external oil phase. Clear phase separation of water-in-oil-in-oil (W/O/O) emulsions will potentially be exhibited due to the hydrophilic nature of Transcutol^®^–PEG mixtures. However, because addition of water and the formation of binary mixtures of Transcutol^®^ with water favor dermal drug delivery, compared to Transcutol^®^ alone it would be inappropriate to include the Transcutol^®^–PEG mixture as the external oil phase and separate water and Transcutol^®^ with an internal oil phase. This may prevent the SDEDDSs from releasing the incorporated drugs, despite exhibiting desirable SDEDDSs properties [12]. Moreover, Transcutol^®^ is considered a well-tolerated topical excipient with the ability to solubilize both hydrophilic and lipophilic drugs, which may be highly beneficial when considering the various lipophilicity profiles of model drugs [12].

### 2.2. Evaluation of the Pseudoternary Phase Diagram to Find Potential Primary Emulsions

A large self-emulsification area was observed, which may signify appropriate excipient combinations. As indicated in Figure 2, four checkpoint formulations were selected as potential primary emulsions (PEs). Checkpoint formulations were selected based on the concept of avoiding regions with poor dermal drug delivery properties. The concentration of the surfactant phase was limited to a ratio of no more than 5 compared to other components of the triplot, because increased surfactant concentrations are known to cause dermal irritation [6]. Moreover, reversed micelle occurrence was avoided by omitting oil ratios > 9 [6]. Lastly, water-rich areas known for micelle formation were avoided by selecting checkpoint formulations of ratios less than 9 in terms of water content. It was consequently decided to select checkpoint formulations with the highest internal oil phase and the highest water content possible due to the capacity of Transcutol^®^ to solubilize both lipophilic and hydrophilic drugs [12]. Furthermore, water is also a crucial component needed to optimize dermal drug delivery facilitated by partnering water and Transcutol^®^, and the inclusion of hydrophilic drugs such as isoniazid (INH) and pyrazinamide (PZY) can benefit from the presence of a greater aqueous component [12].

As indicated in Figure 2, four checkpoint formulations were selected (PE 1, PE 2, PE 3 and PE 4). Next, the PEs were evaluated by being subjected to self-emulsification performance testing and determination of the emulsion stability index (ESI).

### 2.3. Evaluation of Primary Emulsions

#### 2.3.1. Self-Emulsification Performance

Rapid emulsification is highly favorable when developing oral SEDDSs or SDEDDSs, as spontaneous emulsification is a rate-limiting step preceding successful drug absorption [14]. However, drug diffusion through the highly lipophilic SC marks a rate-limiting step during dermal drug delivery [4,6]. Therefore, slower rates of spontaneous emulsification are desired when designing dermal SEDDSs or SDEDDSs, since formulations with elongated self-emulsification rates can predict prolonged contact between the skin and the applied formulations due to the presence of occlusive formulation characteristics [4,6]. Therefore, only formulations that received C or D grading were considered. Fortunately, PE 1, PE 2 and PE 3 received D grading with self-emulsification times of 3 min, 4 min 30 s, and 5 min 10 s, respectively, and were consequently deemed fit for further investigation. However, PE 4 was identified as an E-grade emulsion due to large oil droplet formation, despite a self-emulsification time of 5 min 50 s, only 40 s longer than depicted by PE 3.

#### 2.3.2. Emulsion Stability Index

The PEs were visually inspected after removal from a water bath set at a temperature of 68 °C for a period of 3 h [15]. No phase separation was observed for PE 1, PE 2, or PE 3, which indicated an ESI of 100%. Emulsion instability was found in PE 4 (ESI of 80%), as visible creaming occurred. For this reason, PE 4 was excluded from future characterization experiments.

The poor robustness shown by sudden short exposure to a high temperature may be associated with an increase in the surfactant concentration. Interestingly, an elevation of the surfactant content can reduce the size of the droplets until the surfactant concentration reaches a concentration at which the size of the droplets increases, as established by the aggregates produced from excess surfactant included [16]. The sudden increase in temperature exposure accelerated this instability process, indicating that PE 4 is not suitable for further consideration.

### 2.4. Construction of Pseudoternary Phase Diagrams for Self-Double-Emulsifying Drug Delivery Systems

Figure 3, Figure 4, Figure 5, Figure 6 and Figure 7 indicate the self-double-emulsification areas observed for different combinations of the selected excipients. Pseudoternary phase diagram construction is necessary to confirm if spontaneous emulsification is achievable, and especially to establish the ease and degree of self-emulsification made possible by refined excipient selection during preformulation decision making. Excellent self-emulsification was observed during the addition of natural oils to the fixed mixture of water and Transcutol^®^:PEG together with surfactant. It may seem to be unusual to use the fixed combination of the internal oil phase and the surfactant phase as a single component of the triplot instead of plotting the water and the internal phase together, while plotting the surfactant phase separately. However, the use of decreased surfactant and internal oil phase concentrations are imperative, as the propensity for self-emulsification is intensified with limited surfactant phase content. Decreased surfactant concentrations imply a reduced risk of skin irritation, especially during prolonged use, as well as lowered production costs [4,6]. Moreover, careful attention must be paid to the Transcutol^®^–PEG content of SDEDDSs, as Transcutol^®^ should be included in moderate amounts compared to water, due to its high affinity for water that limits the dermal diffusion if it is not accompanied by sufficient water concentrations [12].

Excellent self-emulsification was exhibited during the construction of the pseudoternary phase diagrams of the tested SDEDDSs, as seen in Figure 3, Figure 4, Figure 5, Figure 6 and Figure 7. Limited variation could be related to the addition of different external oil phases, since the spontaneous emulsification behavior were closely similar However, clear differences can be seen when the region of self-emulsification is compared for different Transcutol^®^:PEG:surfactant phase ratios. This phenomenon is attributed to the instability of SDEDDSs due to increased surfactant concentrations [17]. It should be noted that Transcutol^®^ can also be employed as a co-surfactant [18]. Therefore, if the ratio of Transcutol^®^:PEG:surfactant phase is not optimized, Transcutol^®^ can enhance the effect of the surfactant phase and thus further contribute to the emulsion instability. The gray areas indicated in the 7:4 ratio (internal oil phase:surfactant phase) in the pseudoternary phase diagrams signify that self-emulsification was observed, but the consistency of the SDEDDSs changed to high viscosity, eliciting poor formulation properties such as creaming and excipient precipitation. Again, this is a clear indication that the use of increased concentrations of the surfactant phase are not favorable. Interestingly, this can be related to the reduced difference in the ratio of the internal oil phase compared to the surfactant phase, represented by the 8:3 and 7:4 ratios (internal oil phase to surfactant phase). Therefore, it was decided to maintain an internal oil-phase-to-surfactant-phase ratio of 9:2.

### 2.5. Solubility Studies

Solubility studies are crucial for determining the amount of excipients needed to best solubilize different drug combinations, while maintaining concentrations close to saturation to enhance the driving force needed to achieve dermal diffusion [4,6,19]. The solubility of antitubercular drugs was determined for the following solvents: AVO-, OLV-, EPO-, PALM-, SAFF, and PE, comprising a 9:9:2 ratio (internal oil phase:water phase:surfactant phase), as displayed in Figure 8, Figure 9, Figure 10 and Figure 11. The results are expressed as mean ± SD (*n* = 3).

As indicated, the lipophilic drugs, CFZ and RIF, displayed a higher solubility in natural oils compared to the more hydrophilic drugs, INH and PZY. Moreover, INH and PZY exhibited increased solubility in the PE. As expected, the aqueous solubility of individual drugs indicated in Table 1, increased when solubilized in the PE. This is attributed to the addition of Transcutol^®^ and the surfactant phase to the water phase. This addition improved the aqueous solubility of all drugs, since Transcutol^®^ can solubilize both lipophilic and hydrophilic compounds [12]. Similarly, the inclusion of surfactants is also known to improve the solubility of drugs included in emulsions [20]. Clearly, the PE will be effective in solubilizing INH and PZY. However, an external oil phase must be selected to optimize the solubility of both CFZ and RIF.

PALM induced the highest solubilization of CFZ, whereas RIF had the highest solubility in AVO. However, as CFZ causes dose-dependent skin discoloration, the aim is to reduce the concentration of CFZ when it is used together with PZY. It has been found that the dose of CFZ can be substantially reduced from 25 mg/kg to 6.25 mg/kg when administered orally together with PZY due to the synergistic activity achieved when CFZ and PZY are used in combination [27]. Hence, AVO would be the first choice, as adequate solubility of CFZ would be possible if the CFZ was included in reduced concentrations, while providing a vehicle capable to fully solubilizing RIF. Therefore, AVO would be the ideal external oil phase if CFZ, INH, PZY, and RIF, or CFZ, PZY, and RIF were to be included in a SDEDDS. Nonetheless, if it is decided to include CFZ, INH, and RIF, then PALM or SAFF can also be considered, as increased doses of CFZ will have to be included in the absence of PZY. On the other hand, PALM would be best suited for inclusion of CFZ together with PZY and INH. However, when selecting an external oil phase, the skin penetration enhancement properties of the oil should be considered, as this can greatly affect dermal drug delivery [4,6,11]. In addition, the results obtained during the immiscibility studies should be considered, as a higher degree of immiscibility presents a greater opportunity to successfully formulate SDEDDSs [2,11]. Therefore, for the purpose of this work, AVO was selected as the external oil phase when formulating a SDEDDSs comprising the four antitubercular drugs for microscope examination, due to its favorable solubility properties and the reliable skin penetration enhancement characteristics [6].

### 2.6. Isothermal Micro Calorimetry Compatibility Studies

Isothermal microcalorimetric studies indicated that differences existed between the slopes and y-intercepts of the measured and theoretical (no interaction) heat flows of certain samples tested during compatibility studies. Additionally, the absolute values of the interactions integral to these samples were also greater than zero. Therefore, potential interactions were observed for the following combinations: OLV and PE (−38.01 J/g); AVO and PE (2.64 J/g); EPO and PE (−29.18 J/g); PALM and PE (26.20 J/g); SAFF and PE (85.90 J/g); PE and PZY (−26.54 J/g); PE and CFZ (36.36 J/g); PE and RIF (39.15 J/g); PE and INH (36.61 J/g); CFZ and INH (2252 kJ/g); as well as INH and RIF (822.89 J/g). However, potential interactions between PE and selected external oil phases were expected because the external oil phases were included due to the clear immiscibility between the internal oil phase and external oil phases during the oil immiscibility studies. Therefore, this expected so-called incompatibility is only due to the inability of the oils to mix, and is considered insignificant, as these excipients were intentionally included as a result of their immiscibility with the internal oil phase in order to enable production of double emulsions.

Remarkably, all of the model drugs exhibited potential incompatibility with the PE. Nonetheless, this may be explained by the increased solubility of the drugs observed in the PE compared to the aqueous solubility reported in the literature. The PE comprises Span^®^83, Transcutol^®^, Tween^®^60 and water, excipients that are frequently included in dermal formulations due to their favorable safety, solubilizing, and emulsion-stabilizing properties [4,6,12]. Therefore, this is deemed a physical incompatibility rather than a chemical incompatibility; therefore, it is still possible to use these specific excipient combinations, as physical incompatibilities do not influence formulation stability [6].

In terms of drug–drug interactions, it is noted that synergism is obtained when CFZ is used in combination with the first-line antitubercular agents INH and RIF [28]. This combination is frequently administered to tuberculosis (TB) patients. It has been well documented that RIF shows significant instability in the presence of dissolved INH in an acidic environment [29]. Hence, the emphasis shifts from incompatibility between drugs to the importance of formulation. Selected controlled-release formulations can retain RIF while releasing INH so as to avoid deleterious drug–drug interactions which may lead to reduced drug absorption and decreased therapeutic efficacy [29]. However, as the pH of the skin is normally between 5.4 and 5.9 [4], this is not considered to be sufficiently acidic to exclude the use of a RIF and INH combination in a SDEDDS.

### 2.7. Microscope Examination

Evaluation of the PEs by microscopic examination revealed that the PE ratio of 9:9:2 (internal oil phase:water phase:surfactant phase) was the only PE that exhibited droplet formation. This supports the decision to exclusively use this ratio, and only vary the external oil phase according to the antitubercular drugs included in the SDEDDSs. The microscopic image was captured without any drugs present, as shown in Figure 12.

PE 1 (9:9:2) was subjected to droplet size analysis and polydispersity index (PDI) determination. The results are expressed as mean ± SD (*n* = 3). The droplet size was measured as an average of 173.90 ± 2,30 nm and a PDI value of 0.236 ± 0,11. As shown in Figure 12, the droplet size distribution can be considered homogenous. The accuracy of this ratio is supported by literature comments indicating that the ideal emulsion should contain equal parts water and oil, while being stabilized by appropriate surfactant(s) [30]. However, the presence of PEG as part of the internal oil phase together with Span^®^83 and Tween^®^60 included as surfactants may also contribute to the ability of the PE to remain stable, since PEG can be used as a surface-active agent as well as a solvent [11,31]. Next, PE with a ratio of 9:9:2 was prepared, and AVO was used as the external oil phase due to the favorable solubility properties exhibited by both lipophilic drugs in this vehicle. A checkpoint formulation including a high concentration of the internal oil phase to provide sufficient solubilization of INH and PZY while at the same time providing a moderate amount of the external oil phase, so as to ensure saturated concentrations of the lipophilic components, was selected (Figure 13). This decision is based on the understanding that lipophilic components tend to form a reservoir in the SC or remain in the lipid component of the formulations if the driving force of diffusivity across the SC is not enhanced by formulation techniques such as oversaturation [4,6,11,19].

The checkpoint formulation, namely 9:9:5 (water phase to internal oil and surfactant phase (9:2) to outer oil phase) was prepared. Preparation was commenced by mixing internal oil phase components (Transcutol^®^:PEG ratio of 9:1) by means of gentle stirring until fully combined. At the same time, the surfactant phase was separately prepared by gently stirring Span^®^60 and Tween^®^83 in a ratio of 1:1 until completely blended. Next, surfactant phase and the internal oil phase were combined (9:2 ratio of internal oil phase to surfactant phase) while being subjected to mild stirring for a duration of 30 min. Thereafter, a predetermined quantity of water, a ratio of 9 in this case, was added in small increments while applying steady agitation. After subjecting this mixture to homogenization for 4 min, in order to generate a PE, the pre-determined quantity of five parts external oil phase was added in a dropwise fashion while gently stirring. After continuous stirring, the mixture was removed from the stirring plate and subjected to sonication until the desired droplet size was achieved. Drugs were included at a concentration of 2% each. Hence, overall composition of SDEDDS can be summarized as follows: 8% drugs (2% of each drug), 36% water, 20% AVO, 29.5% internal oil phase, and 6.5% surfactant phase.

The checkpoint SDEDDS was characterized by measuring droplet size and PDI. The results are expressed as mean ± SD *(n = 3)*. Average droplet size was revealed to be 92.46 ± 5.52 nm, and the PDI value found to be 0.391 ± 0.05. Therefore, the selected PE and SDEDDS were visually examined via transmission electron microscopy (TEM) for the purpose of gaining a reliable visual representation of the droplets that fell into the nano size range. As the highlight of this study, the TEM observation revealed clear small droplet formation inside larger smooth-surfaced droplets, signifying the successful development of a SDEDDS designed to simultaneously contain four drugs. Moreover, the mechanism of spontaneous emulsification was captured as indicated in Figure 14.

As indicated in Figure 14, clear formation of a small droplet(s) within a larger droplet was achieved. Interestingly, a membrane structure is established by the surfactant phase (A), and then water is engulfed by the surfactant phase, to be captured as small droplets inside the internal oil phase (B and C). However, when considering the scale bar of 100 nm on the TEM image, it is evident that the average droplet size measured during characterization experiments only measured the larger droplets harboring the smaller droplets (D). Hence, this may be considered a novel contribution to the development of dermal SDEDDS, since nano-sized droplets are known to cross the SC with ease compared to larger-sized droplets. The opportunities of delivering fixed-dose drug combinations via dermal application when included in SDEDDSs instead of SEDDSs can be considered when studying the TEM images of the PE (SEDDS) compared to SDEDDS, displayed in Figure 15a,b.

As seen in Figure 15a,b, SEDDS is presented as the (a) PE included in (b) SDEDDS. The most noticeable difference is in droplet size, since (a) rendered an average droplet size of 173.90 nm and (b) depicted an average droplet size of 92.46 nm. Interestingly, PDI values signify that (a) is more homogenous compared to (b). However, upon visual examination, (b) seems to be a sample of increased homogeneity compared to (a). This can be attributed to the observation made previously that only large droplets of SDEDDS were measured during droplet size determination due to the incorporation of smaller droplets inside the larger droplets. However, in terms of drug delivery, the TEM images indicate multiple phases presented by SDEDDSs compared to SEDDSs are able to incorporate more than one drug. This is due to the different phases established by water and two immiscible oils than can solubilize polar and non-polar drugs in the same vehicle with potential sustained release characteristics. This aspect requires further examination. TEM images were captured in the absence of drugs, since the red discoloration caused by both CFZ and RIF limits the contrast required for identification of SEDDS and SDEDDS, respectively. Therefore, the importance of preformulation studies during the development of SDEDDSs is emphasized and can be considered the key to success. This is indeed a novel development. To the knowledge of the authors, this represents the first nano-SDEDDS (N-SDEDDS) or W/O/O self-nano-emulsifying drug delivery system (SNEDDS) capable of incorporating four individual drugs that possess strikingly different physicochemical properties.

## 3. Materials and Methods

### 3.1. Materials

Combinations of CFZ, INH, PZY, and RIF were utilized as model drugs. CFZ, INH, and RIF were kind gifts from Prof Wilna Liebenberg, head of the Solid-State Pharmaceutical Innovation and Nanotechnology (SPIN) research group at the North-West University (Potchefstroom, South Africa). PZY, Transcutol^®^, Span^®^83, and Tween^®^60 were purchased from Merck (Darmstadt, Germany). AVO, beeswax, EPO, linseed oil, OLV, PALM, SAFF, and shea butter were bought from Nautica Organics Trading (Durban, South Africa). PEG was obtained from DB Fine Chemicals (Sandton, South Africa). Distilled water was acquired through a Rephile Bioscience Ltd. system (Boston, MA, USA).

### 3.2. Methods for Drug and Excipient Selection

#### Considering Drug Selection

Antitubercular drugs were selected on the basis of their physicochemical properties and Biopharmaceutical Classification System (BCS) ranking, as indicated in Table 1, to demonstrate the capacity of SDEDDSs to incorporate multiple drugs of different physicochemical natures.

From Table 1, the lipophilicity of drugs may be classified as CFZ >>> RIF >> INH >> PZY. These physicochemical properties were considered in order to obtain an exceptional dermal drug delivery capacity for the SDEDDSs, as the optimal lipophilicity of drugs destined for dermal drug delivery is established at a Log P value of 1–3 and a molecular weight <500 Da [32,33,34]. None of the selected drugs has a Log P value that falls within the prescribed range for desirable dermal drug delivery. Fortunately, though, INH and PZY have acceptable molecular weights. CFZ, on the other hand, presents a compliable molecular weight. However, this drug is highly lipophilic, as can be concluded by its Log P value of 7.66 and elimination half-life of 70 days. Lastly, the molecular weight of RIF drastically exceeds the prescribed molecular weight, and thus renders it unsuitable for dermal drug delivery. Nevertheless, a recent publication indicated that drugs with a molecular weight that does not exceed 1000 Da can be considered acceptable for dermal drug delivery [35]. Consequently, lipophilicity is considered the determining factor that prohibits the delivery of dermal drugs, instead of the molecular weight of the drugs chosen for this study [35,36].

Developing a topical dosage form containing four drugs to aid in cutaneous tuberculosis (CTB) disease is challenging but necessary due to the daunting nature of *Mycobacterium tuberculosis* [19]. Hence, the requirements for an effective tuberculosis regimen comprise simultaneous administration of various bactericidal and sterilizing anti-tubercular agents that are administered for a sufficient period for the purpose of sustaining anti-microbial effectiveness, while preventing mutations of *Mycobacterium tuberculosis* that might lead to drug resistance [19,37]. Hence, drug combinations are the norm for anti-tubercular treatment regimens [37]. The first-line anti-tubercular drugs utilized during this study included INH, PZY, and RIF [19]. However, resistance against first-line agents is drastically increasing, and there is a need for new anti-tubercular drug combinations as well as new drug entities to assist in the fight against TB [37,38].

CFZ, a riminophenazine antibiotic, is listed by the World Health Organization as a Category B agent for treatment of multidrug-resistant TB (MDR-TB) and extensively drug-resistant TB (XDR-TB) [39]. A meta-analysis study discovered that the overall pooled proportion of treatment success was 61.96% after providing CFZ treatment, compared to a global success rate of 54% for MDR-TB patients and a limited 30% success for XDR-TB patients [39,40,41]. Additionally, a prospective, randomized, multicenter study concluded that MDR-TB patients treated with the shorter regimen including CFZ had a comparable successful outcome rate when compared to patients treated with the standard regimen [39,42]. Moreover, synergistic effects were observed when utilizing CFZ in combination with first-line anti-tubercular agents in both planktonic and biofilm-forming cultures [43]. Therefore, potential improvements in therapeutic efficacy may be achieved by adding CFZ to standard, readily available anti-tubercular treatment regimens [43]. Thus, CFZ is a worthwhile candidate to consider for treatment of CTB, since CTB is currently treated with first-line anti-tubercular agents [43]. However, SDEDDSs developed during this study can also aid in treatment of non-tuberculosis mycobacteria (NTM) skin infections, since NTM is known for its intrinsic, inducible, and adaptive resistance mechanisms [44]. Therefore, treatment periods of 2–4 months are required for NTM skin and soft tissue infections accompanied by co-administration of multiple antibiotics [44]. CFZ is recommended for treatment of *Mybobacterium fortuitum* and *Mycobacterium abscessus* complex together with other agents such as amikacin, trimethoprim–sulfamethoxazole, linezolid, tetracyclines, quinolones, and gepotidacin (for *Mycobacterium fortuitum* complex) and tigecycline, clarithromycin, omadacycline, and thiostrepton (for *Mycobacterium abcessus* complex) [44]. Unfortunately, the co-administration of antibiotics can lead to challenges such as drug interactions, drug-related adverse reactions, and high medication costs that have the potential to compromise treatment options as well as patient compliance [44]. However, SDEDDS can be a valuable tool during the development of individualized therapy for conditions such as NTM skin infections and CTB, since multiple drugs as well as different drug combinations can be added to SDEDDSs that are easy to manufacture and upscale if excipient selection is performed correctly [19].

### 3.3. Preformulation Studies

#### 3.3.1. Oil Immiscibility Studies

It was decided to formulate a W/O/O emulsion for the purpose of providing a water phase to optimally solubilize the hydrophilic drugs, and oil phases capable of ensuring effective solubilization of the lipophilic drugs. Ideally, the external oil phase should possess additional skin penetration enhancement properties to facilitate dermal drug delivery [45]. However, the art of creating an oil-in-oil (O/O) emulsion lies in the mixing of two immiscible lipophilic components to maintain the separation of an internal and external oil phase [2,46]. Therefore, it would be best to employ a ‘lipophilic oil’ as well as a more ’hydrophilic oil’ into a single SDEDDS. Thus, the hydrophilic–lipophilic balance (HLB) values of oils were studied to find oils or excipient mixtures with low and elevated HLB values in order to create and maintain internal and external oil phase separation.

Beeswax (HLB value of 4 or 12), linseed oil (HLB value of 3.23) and Transcutol^®^:PEG (9:1 ratio), with a combined HLB value of 4.46, were analyzed to find an internal oil phase [47,48,49]. Shea butter, AVO, OLV, EPO, and PALM were considered as potential external oil phases. Immiscibility experiments were performed by weighing 10 g of a potential internal oil phase and adding it to 10 g of a tested external oil phase [2]. Next, the oil mixtures were subjected to vortexing for 10 min [2]. Thereafter, the vortexed mixtures were poured into separatory funnels and left for 30 min in order to evaluate the degree of separation of the different oil phases [2]. This would enable the determination of the immiscibility percentage in terms of weight differences [2].

#### 3.3.2. Hydrophilic–Lipophilic Balance Consideration

Choosing the correct surface active agent(s) is essential in order to stabilize the emulsions once formulated [50]. The HLB system allows theoretical quantification of the ability of a surfactant and co-surfactant combination to stabilize an emulsion [50,51,52]. When considering the formulation of multiple emulsions such as SDEDDSs, the inclusion of at least two surface active agents is required to stabilize the formation of both primary and secondary emulsions that as a whole present a multiple emulsion [53]. In the case of W/O/O emulsions, one surface active agent should have a low HLB value to stabilize the formation of a PE at the water-in-oil (W/O) interface, and a second surface active agent with an enhanced HLB value should provide stabilization of the secondary emulsion formation [53]. If surface active agents are selected correctly, the stabilization established by these excipients form the building blocks needed by SDEDDSs to provide protection of the incorporated drugs, the ability to include several drugs due to different emulsion compartments, and to enable sustained release from multi-emulsions [53]. Therefore, the selection of surface active agents for multiple emulsions cannot be based solely on the same concept applied for simplified emulsions. For the latter, a non-ionic, lipid-soluble surfactant with an HLB value ranging between 3–8 will tend to deliver O/O emulsions. A non-ionic surfactant that is preferentially water-soluble, has an HLB value around 8–16, and will tend to stabilize O/W emulsions [54]. Therefore, for the purpose of this study, Span^®^83 (HLB value of 3.7) and Tween^®^60 (HLB value of 15) were selected as surfactant and co-surfactant, respectively [55,56]. A fixed ratio of Span^®^83:Tween^®^60 (1:1) was maintained throughout the experiments, as this ratio has demonstrated greater stabilization of SEDDSs compared to higher ratios that increase the emulsion range but reduce stability, as indicated by drug precipitation [6,57].

#### 3.3.3. Construction of a Pseudoternary Phase Diagram to Find Potential Primary Emulsions

A water titration method was utilized to construct a pseudoternary phase diagram wherein Transcutol^®^:PEG (9:1), water and the surfactant phase (Span^®^83:Tween^®^60; 1:1) were combined in different ratios to display a triplot. This then is able to indicate the most appropriate combination of additives required to form successful SDEDDSs, as one can establish the self-emulsifying areas and also validate the most effective concentrations and ratios of the included composites. No drugs were used, as it was necessary to establish excipient behavior without the influence of any drugs. The surfactant phase and water were mixed in fixed ratios (10:0, 9:1, 8:2, 7:3, 6:4, 5:5, 4:6, 3:7, 2:8, 1:9, 0:10), while Transcutol^®^:PEG (9:1) was added in a dropwise manner as a variable component at an ambient temperature. The concentration at which the turbidity of the mixtures is first observed was plotted as the endpoint, employing Triplot software version 4.1.2 [6,58]. Thus, the area of self-emulsification was established as required to find potential PEs.

#### 3.3.4. Evaluation of Primary Emulsions

The PEs selected from the pseudoternary phase diagram were subsequently prepared by adding known quantities of the internal oil phase (Transcutol^®^:PG; 9:1) to established quantities of the water and surfactant phases. The mixtures were then subjected to homogenization with a Heidolph DIAX600 homogenizer (Schwabach, Germany) for 4 min at 9500 rpm, while being kept at room temperature [2,59]. The PEs were evaluated after a waiting period of 24 h at an ambient temperature in terms of emulsion stability index and self-emulsification performance in order to eliminate poor excipient combinations from further experimentation.

#### 3.3.5. Self-Emulsification Performance

Complete emulsification and emulsification time can be defined as the time required to obtain a macroscopically homogeneous emulsion [60]. To obtain self-emulsification times, PEs were assessed using a Type II Distek 2500 dissolution apparatus (2501049, North Brunswick, NJ, USA). A sample (1 mL) was collected from individual PE formulations and diluted with 100 mL distilled water and maintained at a fixed temperature of 32 ± 0.5 °C with a paddle rotation speed set at 50 rpm to provide gentle agitation. The time it takes for PEs to form a homogeneous mixture upon dilution was noted and classified according to the following grading system [61]:Grade A: Fast-forming emulsion (within 1 min) with a clear or bluish color;Grade B: Fast-forming (within 1 min) and somewhat less clear emulsion with a bluish-white color;Grade C: Fine murky emulsion that forms within 2 min;Grade D: Dull, grayish-white emulsion that displays a slightly oily appearance, which indicates slow emulsification (longer than 2 min);Grade E: A formulation demonstrating either poor or minimal emulsification with large oil droplets existing on the surface.

Emulsions graded as C or grade D were deemed favorable for dermal drug delivery [4]. For this reason, all emulsions that received a grade other than grade C or D were deemed unsuitable for further investigation.

#### 3.3.6. Emulsion Stability Index (ESI)

The PEs (10 mL) were placed in a water bath set at a temperature of 68 °C for a period of 3 h prior to evaluation [2]. After the 3 h waiting period, the PEs were inspected to determine the variance between the separated layer compared to the initial total volume of the emulsion placed in the water bath [2]. This provides a clear indication of the capacity of an emulsion to retain stability once it has been exposed to increased temperatures for a short time. The volume of the separated layer was compared with the total volume of the initial emulsion to calculate the ESI using the following equation:(1)$$ESI=1-\left(\frac{Volume\;of\;separated\;layer}{Total\;volume\;of\;emulsions}\right)\times 100\%$$

#### 3.3.7. Construction of Pseudoternary Phase Diagrams for Self-Double-Emulsifying Drug Delivery Systems

The water titration method was again utilized to construct pseudoternary phase diagrams. Drugs were again excluded during pseudoternary phase diagram construction to be able to observe excipient response without the complication of variables introduced by drug inclusion. The internal oil phase was mixed together with the surfactant phase (i.e., PE 1, PE 2 and PE 3, respectively) and water in fixed proportions (9:1, 8:2, 7:3, 6:4, 5:5, 4:6, 3:7, 2:8 and 1:9), while individual external oil phases were added dropwise as the variable component at an ambient temperature. The concentration at which the turbidity of the mixtures is observed was plotted as the endpoint, employing Triplot software version 4.1.2 [6,58,62,63].

#### 3.3.8. Solubility Studies

An excess amount of individual antitubercular drugs was added to 5 mL of the different vehicles considered during this study. Samples were then vortexed for approximately 2 min. Solubility studies were performed by placing samples in the circular axis (54 rpm) of a rotating solubility bath set at 32 °C (±0.5 °C) for a period of 48 h [6]. The samples were then centrifuged at 3000 rpm for 15 min, followed by the removal of the supernatant from each sample [6]. The supernatant was filtered through a 0.45 µm Millipore^®^ filter and diluted with methanol [6]. A validated high-performance liquid chromatographic method was employed to analyze all samples [6].

#### 3.3.9. Isothermal Micro Calorimetry Compatibility Studies

The compatibility of excipients and drugs was investigated using a previously published method [64]. A 2277 Thermal Activity Monitor, TAM III (TA Instruments, New Castle, DE, USA), equipped with an oil bath with a stability of ±100 µK over 24 h was used. The temperature was maintained at 40 °C and 100 mg samples were tested. The heat flow for each individual component was measured to generate a theoretical response (i.e., baseline), which was accompanied by a comparison of the theoretical response with the calometric output to determine compatibility. If the theoretical response drastically differed from the detected calometric output, interactions between excipients were suspected.

#### 3.3.10. Microscope Examination

All visual examinations of either the PEs or the SDEDDSs were conducted utilizing an Olympus microscope (Tokyo, Japan). TEM measurements were performed on a TECNAI G2 (ACI) instrument operated at an accelerating voltage of 200 kV (Hillsboro, OR, United States of America) [65,66].

#### 3.3.11. Droplet Size and Size Distribution

Droplet size and size distribution were assessed by means of dynamic light scattering performed using a Malvern Zetasizer Nano^®^ ZS (Worcestershire, UK) at 25 °C [6]. All samples were analyzed in triplicate.

## 4. Conclusions

Future in vitro drug release and permeation studies should be conducted in order to establish the release of the drugs from SDEDDSs. Additionally, nanotoxicology studies should also be performed for the purpose of establishing the extent as well as the safety of the permeation achieved by N-SDEDDSs [67]. However, the final SDEDDS presented an average droplet size of 92.46 nm and a PDI value of 0.391 which is considered exceptionally small on the basis of literature data [68,69,70]. Therefore, dermal drug delivery can be considered very likely due to the small droplet size of the SDEDDS [68,69,70]. This study has clearly demonstrated the potential of SDEDDSs as a dermal drug delivery vehicle by including four model drugs in a single SDEDDS. This was challenging, since the model drugs possess very different physicochemical properties, with individual aqueous solubilities ranging from sparingly soluble to a solubility of 125 mg/mL. Moreover, the selected drugs varied in terms of their BCS classification, namely Class I, II, and III. However, the development of the SDEDDS described here was based on findings from the preformulation studies. Therefore, this work signifies an important step in conducting thorough preformulation experiments that provide a foundation for the formulation and subsequent characterization of SDEDDSs tailored to accommodate selected drugs while being adapted for dermal drug delivery. Furthermore, this paper indicates the importance of carefully constructing drug delivery systems from start to finish—preformulation studies represent the key to success for development of dermal SDEDDSs. To the authors’ knowledge, this represents the first published study indicating that SDEDDSs can incorporate four individual drugs successfully into a single dosage form.

## Figures and Tables

**Figure 1 pharmaceuticals-16-01348-f001:**
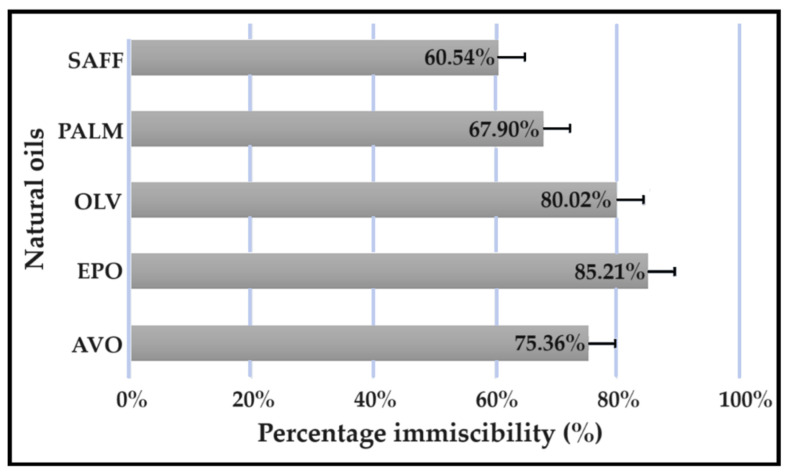
Immiscibility percentage of individual natural oils mixed with a combination of Transcutol^®^:PEG (9:1).

**Figure 2 pharmaceuticals-16-01348-f002:**
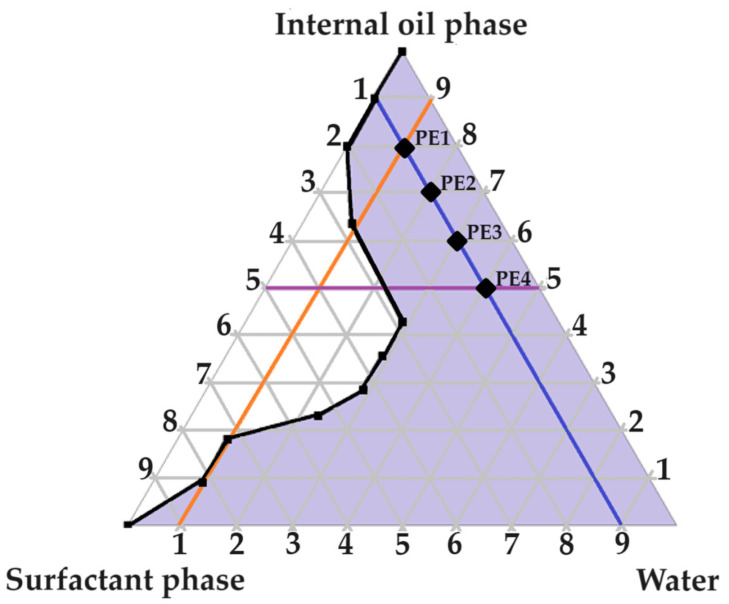
Pseudoternary phase diagram of water, surfactant phase and internal oil phase. The numbering represents the given parts of the different phases included in a ratio at a specific point on the pseudoternary phase diagram.

**Figure 3 pharmaceuticals-16-01348-f003:**
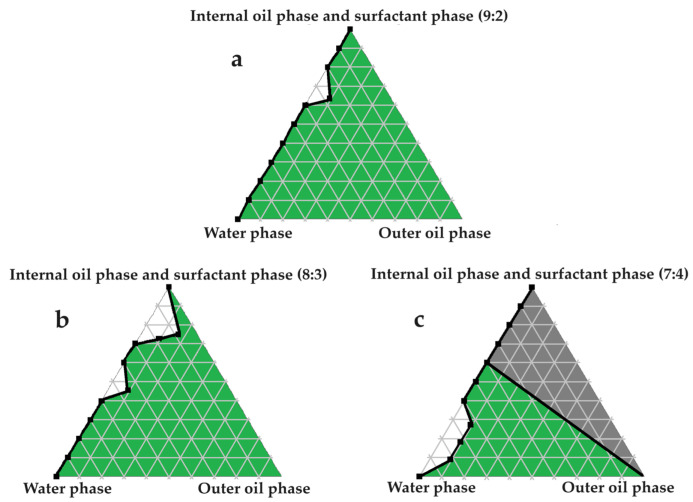
Pseudoternary phase diagrams of water, avocado oil, and fixed proportions of the surfactant and internal oil phases. In (**a**) the internal oil and surfactant phases are combined in a ratio of 9:2; in (**b**) the internal oil and surfactant phases are blended in an 8:3 ratio; and in (**c**) the internal oil and surfactant phases are included as a 7:4 ratio.

**Figure 4 pharmaceuticals-16-01348-f004:**
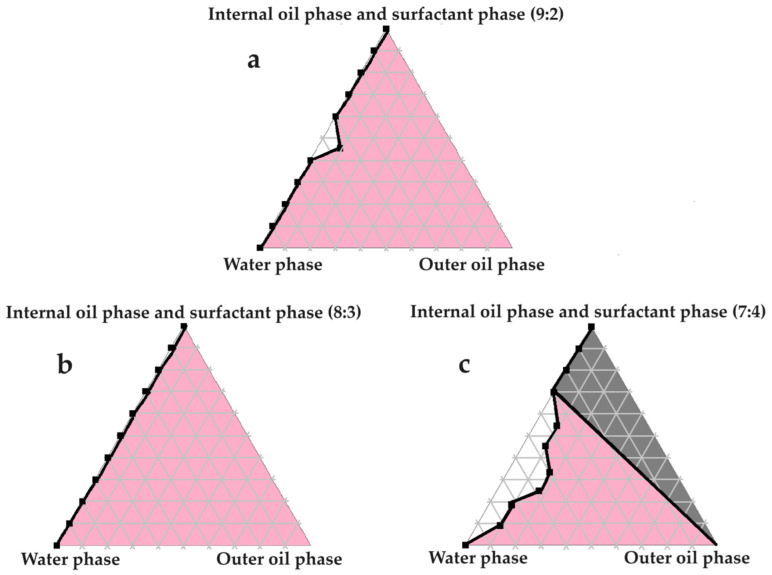
Pseudoternary phase diagrams of water, evening primrose oil, and secure fractions of the surfactant and internal oil phases. In (**a**) the internal oil and surfactant phases consist of a 9:2 ratio; in (**b**) the internal oil and surfactant phases comprise an 8:3 ratio; and in (**c**) the internal oil and surfactant phases are combined in a 7:4 ratio.

**Figure 5 pharmaceuticals-16-01348-f005:**
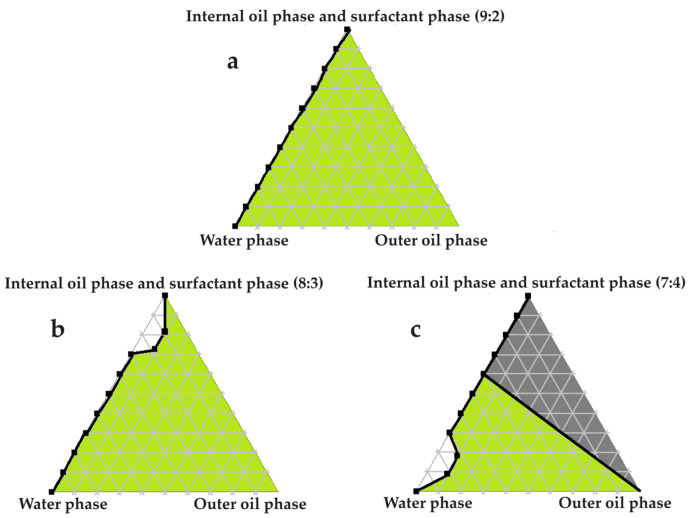
Pseudoternary phase diagrams of water, olive oil, and fixed ratios of the selected surfactant and internal oil phases. In (**a**) the internal oil and surfactant phases comprise a ratio of 9:2; in (**b**) the internal oil and surfactant phases consist of an 8:3 ratio; and in (**c**) the internal oil and surfactant phases are blended in a 7:4 ratio.

**Figure 6 pharmaceuticals-16-01348-f006:**
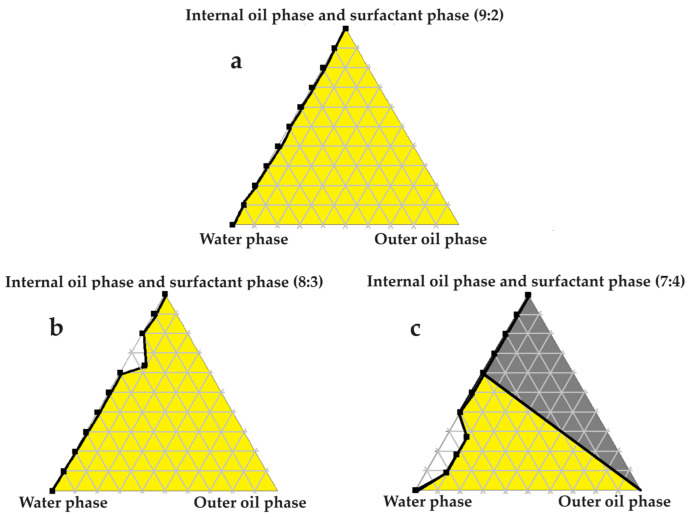
Pseudoternary phase diagrams of palm oil and set ratios of the chosen surfactant and internal oil phases. In (**a**) the internal oil and surfactant phases are combined in a ratio of 9:2; in (**b**) the internal oil and surfactant phases are blended in an 8:3 ratio; and in (**c**) the internal oil and surfactant phases are included as a 7:4 ratio.

**Figure 7 pharmaceuticals-16-01348-f007:**
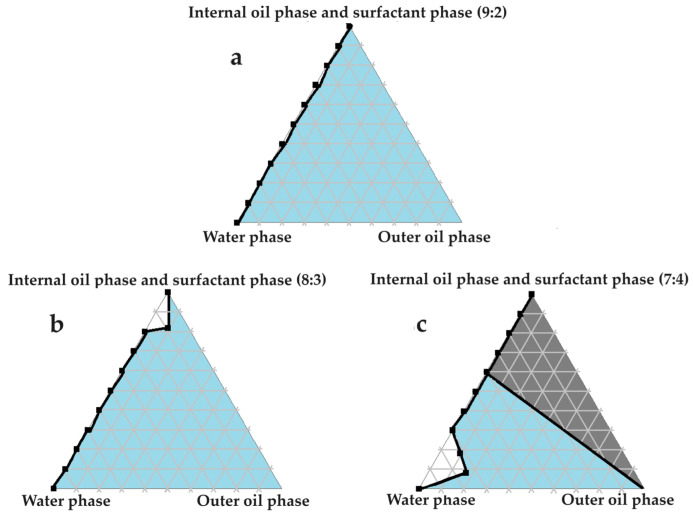
Pseudoternary phase diagrams of water, safflower oil, and set proportions of the surfactant phase as well as the internal oil phase. In (**a**) the internal oil and surfactant phases consist of a 9:2 ratio; in (**b**) the internal oil and surfactant phases comprise an 8:3 ratio; and in (**c**) the internal oil and surfactant phases are combined in a 7:4 ratio.

**Figure 8 pharmaceuticals-16-01348-f008:**
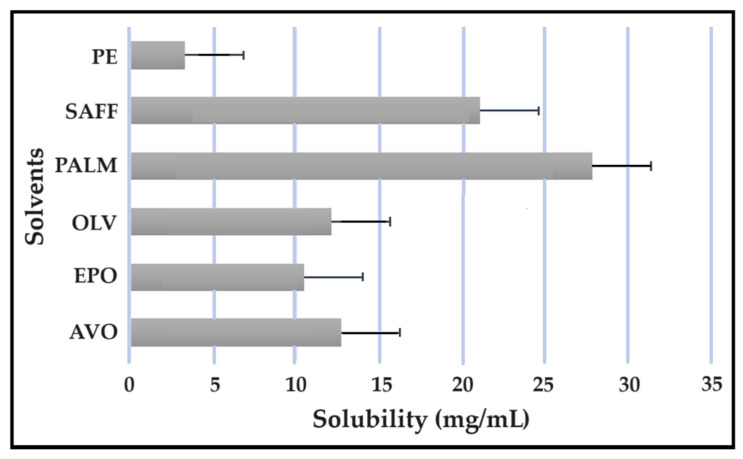
Solubility of CFZ in the solvents.

**Figure 9 pharmaceuticals-16-01348-f009:**
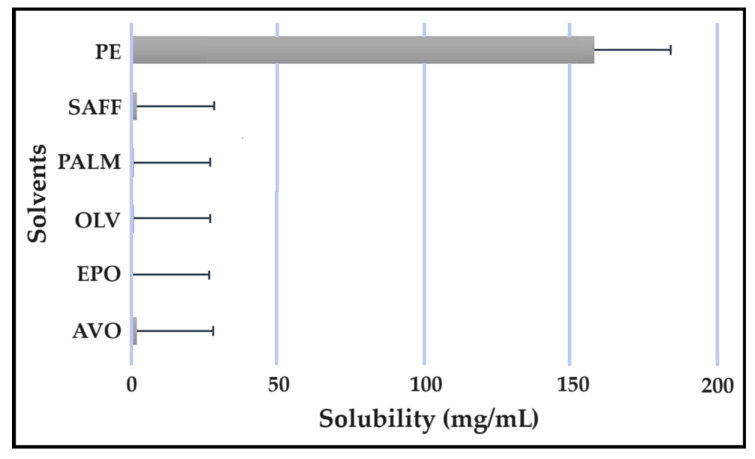
Solubility of INH in the solvents.

**Figure 10 pharmaceuticals-16-01348-f010:**
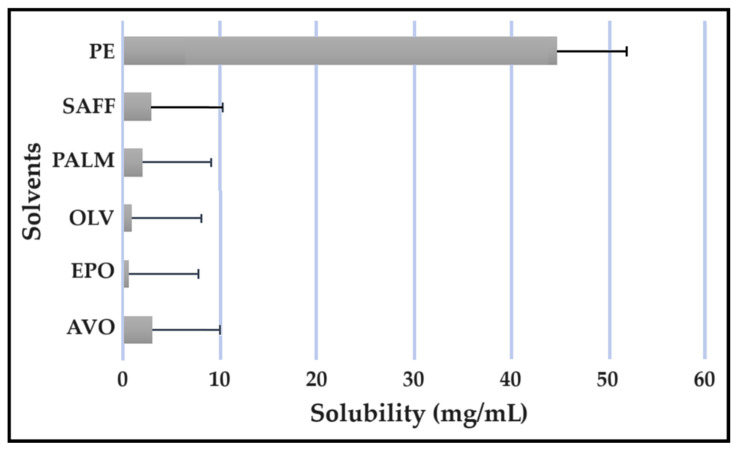
Solubility of PZY in the solvents.

**Figure 11 pharmaceuticals-16-01348-f011:**
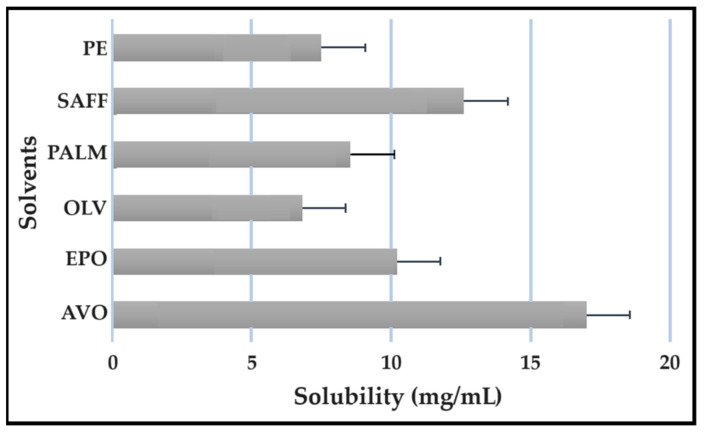
Solubility of PZY in the solvents.

**Figure 12 pharmaceuticals-16-01348-f012:**
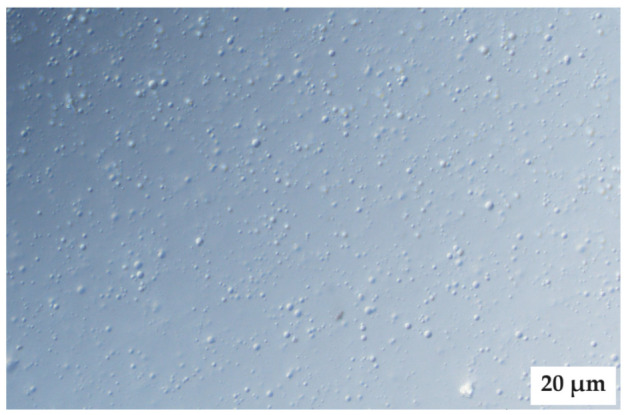
Microscopic image of PE 1 (9:9:2 ratio) comprising no active ingredients.

**Figure 13 pharmaceuticals-16-01348-f013:**
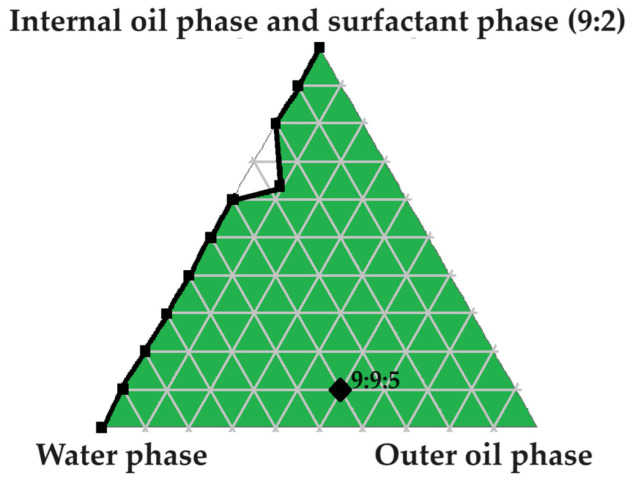
Pseudoternary phase diagram utilized to select a checkpoint formulation (9:9:5) for SDEDDS development.

**Figure 14 pharmaceuticals-16-01348-f014:**
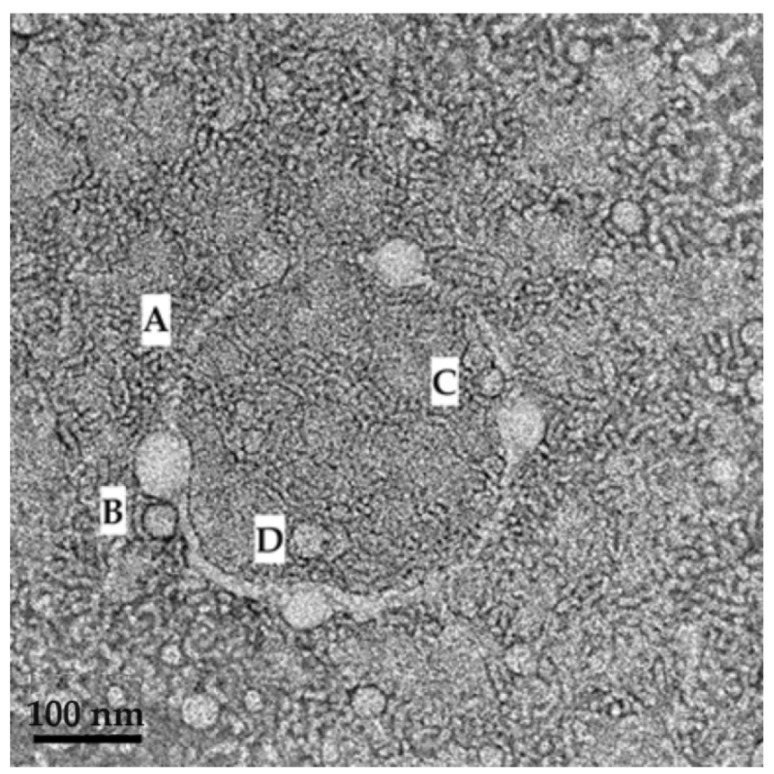
TEM image revealing the mechanism of SDEDDS formation where (**A**) a membrane structure is established by the surfactant phase; (**B**,**C**) and then water is engulfed by the surfactant phase so as to form small droplets inside the internal oil phase. (**D**) indicates where larger droplets harbor smaller droplets inside.

**Figure 15 pharmaceuticals-16-01348-f015:**
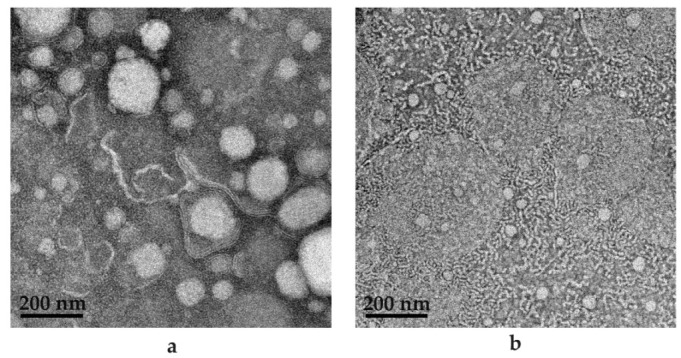
TEM images for (**a**) PE (9:9:2) comprising water to internal oil phase to surfactant phase; (**b**) SDEDDS (9:9:5) containing water to internal oil phase and surfactant phase (9:2) to external oil phase.

**Table 1 pharmaceuticals-16-01348-t001:** Physicochemical properties of the selected anti-tubercular drugs [6,19,21,22,23,24,25,26,27,28,29,30,31].

Drug	Log P	Molecular Weight (D_a_)	Aqueous Solubility	Elimination Half-Life	Metabolized	BCS Classification
CFZ	7.66	473.40	<1 mg/mL	70 days	Hepatic	Class II
INH	0.64	137.14	125 mg/mL	45–110 min ^1^ 2–4.5 h ^2^	Hepatic	Class I/III ^3^
PZY	−1.88	123.11	15 mg/mL	3–5 h	Hepatic	Class III
RIF	3.80	822.90	1.51 mg/mL	3–4 h	Hepatic	Class II

^1^ rapid metabolizers. ^2^ slow metabolizers. ^3^ INH is considered highly water soluble; however, data that reflect its oral absorption and permeability are inconclusive. Therefore, we suggest that INH can be regarded as somewhere between BCS Class I and III.

## Data Availability

Data is contained within the article.

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
