# Peer review of "The Development of Dermal Self-Double-Emulsifying Drug Delivery Systems: Preformulation Studies as the Keys to Success"

_pharmaceuticals, 2023, doi:10.3390/ph16101348_

Round 1
Reviewer 1 Report
Dear Author,
I would like to inform you that the research study entitled “Development of Dermal Self-Double-Emulsifying Drug Delivery Systems: Preformulation Studies the Keys to Success” has been intensively reviewed and evaluated. The rationale of this study is outstanding but there were some major points that need to be intensively and carefully revised. After the completion of them, it could be a good candidate for literature.
Hereby I would like to present my comments:
Comment_1: Please indicate that the rationale of providing 4 drugs in a dosage form (It could be outstanding, but is there a combinatory treatment regimen? If so, please explain pharmacologically (doses, examples…etc.). May be a pharmacological subheading or paragraph helpful.
Comment_2: Please select high-resolution images for the figures (Fig.3-7). If it is not possible, please explain each of them in the figure captions.
Comment_3: Please indicate the scale bars on the microscopy images.
Comment_4: The characterization part of the manuscript was found to be a little bit weak. Thus, droplet/particle size and poly dispersity index of formulations must be shown. Additionally, the chemical interactions or compatibility should be proven by using FTIR.
Comment_5: As an integral part of preformulation studies, there is a need for in vitro drug release studies to observe release behavior from the formulations. The release kinetics should also be discussed for dermal applicaltions.
Comment_6: Abbreviations should clearly be explained, please check all of them (Is PE primary emulsion?).
Comment_7: L478 Brand name and country of high-pressure homogenizer should be written.
Comment_8: Please check the line 715 is there a reference missing or the “56” is misused?
Best regards.
Author Response
Please find the rebuttal attached.
Kind regards,

Reviewer 2 Report
pharmaceuticals-2565260
Development of Dermal Self-Double-Emulsifying Drug Delivery Systems: Preformulation Studies the Keys to Success
The manuscript by Staden et al. described the development of Self-Double-Emulsifying Drug Delivery Systems to incorporate four different drugs in dermal delivery. The authors presented some data for the pre-formulation steps. They seem to be sufficient for the conclusion. However, the manuscript still needs to be improved. Below are some recommendations for revising this manuscript.
1. The particle size and size distribution of emulsions (by DLS) are critical parameters for evaluating SDEDDSs. The authors need to include them in the manuscript.
2. Figures should be improved. The style should be consistent. Many figures are of low quality.
3. Figures 12 and 14: Microscopic images are of poor quality. From Figure 14, the emulsion seems to be not homogenous. What are smaller/ tiny droplets? Please show the scale to specify the relative size of droplets. The authors need to show more emulsion droplets in the image to conclude “The checkpoint formulation was prepared and subjected to microscopic examination as shown in Figure 14. As the highlight of this study, microscopic observation revealed clear, small droplet formation inside larger, smooth-surfaced droplets, signifying the successful development of a SDEDDS containing four drugs simultaneously”. Please provide other images.
4. Figure 14 legend: “Microscope image of the formulated SDEDDS containing CFZ, INH, PZY, and RIF”. So, does this formulation contain 4 drugs? Please mention the drugs and SDEDDS compositions/ amounts.
5. To verify the “formation of a small droplet(s) within a larger droplet was achieved” as stated by the authors, the authors should provide an image of emulsions from SEDDS to compare. From 2 sets of images, we can easily identify the effects of the additional oils in SDEDDS.
6. There is no description of the method to prepare drug-loaded SDEDDS. Please provide detailed component ratios or amounts.
7. ESI was mentioned in the method section but seemed not to be used. Please provide relevant data.
8. “Self-emulsification performance” was mentioned in the method section but seemed not to be used. Please provide relevant data.
9. Drug release and permeation should be included to show the applicability to dermal drug delivery.
Minor editing of English language required
Author Response

(The authors gave the same response as above.)

Round 2
Reviewer 1 Report
Dear Author,
Following a thorough examination, the responses you have provided in response to the requested revisions have been deemed satisfactory. Consequently, I am inclined to express a positive opinion regarding the publication of your manuscript. I extend my best wishes for your continued success.
Best regards
Author Response
Dear Reviewer,
Thank you kindly for your remarks. We wish you all the best.
Kind regards
Reviewer 2 Report
pharmaceuticals-2565260
The manuscript was revised, and some issues were resolved. However, the manuscript is still unsuitable for publication. Please see the details below to improve it.
1. The particle size and PDI should be presented as means ± SDs with n≥3. Relevant methods should be included in the Method section.
2. The Figures' quality is not improved and is unsuitable for publication.
3. “It is true that drug release studies and dermal diffusion studies are the final goal of this work. However, it is intended for a separate publication. Hence, the authors have mentioned the importance of in vitro release and permeation studies as part of the future prospects of this work.” => The data should be included since they are critical to evaluate the formulations.
4. “Drugs were included at a concentration of 2% each” => This is unclear (2% of which, please specify the drug amount/ ratio to SMEDDS components).
5. From TEM and SEM, it may be hard to conclude anything regarding the structures of the particles. Since the authors stated that “clear formation of a small droplet(s) within a larger droplet was achieved” (lines 383 – 384), I recommend adding some more discussions as well as markers on the TEM to help readers further comprehend the statement.
Author Response
Please find attached.
Kind regards,

Round 3
Reviewer 2 Report
The manuscript was partly improved. Please consider some points below.
1. Please check and use "." as decimal separators.
2. The figure quality is still low. But I will leave this for the Editors to decide.